# *Elsholtzia ciliata* (Thunb.) Hyland: A Review of Phytochemistry and Pharmacology

**DOI:** 10.3390/molecules27196411

**Published:** 2022-09-28

**Authors:** Fulin Wang, Xue Liu, Yueru Chen, Ying An, Wei Zhao, Lu Wang, Jinli Tian, Degang Kong, Yang Xu, Yahui Ba, Honglei Zhou

**Affiliations:** College of Pharmacy, Shandong University of Traditional Chinese Medicine, Jinan 250355, China

**Keywords:** *Elsholtzia ciliata* (Thunb.) Hyland, phytochemistry composition, pharmacological activities

## Abstract

In this paper, the confusion of the sources of medicinal materials was briefly expounded, and the differences among the varieties were pointed out. At the same time, the chemical components and pharmacological properties of *Elsholtzia ciliata* (Thunb.) Hyland (*E. ciliata*) were reviewed. The structures of 352 compounds that have been identified are listed. These mainly include flavonoids, terpenoids, phenylpropanoids, alkaloids, and other chemical components. They have antioxidant, anti-inflammatory, antimicrobial, insecticidal, antiviral, hypolipidemic, hypoglycemic, analgesic, antiarrhythmic, antitumor, antiacetylcholinesterase, and immunoregulator activities. At present, there are many researches using essential oil and alcohol extract, and the researches on antioxidant, anti-inflammatory, anti-microbial, and other pharmacological activities are relatively mature. This paper aims to summarize the existing research, update the research progress regarding the phytochemicals and pharmacology of *E. ciliate*, and to provide convenience for subsequent research.

## 1. Introduction

*Elsholtzia ciliata* (Thunb.) Hyland belongs to the genus *Elsholtzia*, family Lamiaceae. In the clinical application of traditional Chinese medicine, the aerial parts of *Mosla chinensis* Maxim (MCM) and *Mosla chinensis* Maxim cv. Jiangxiangru (JXR) are used as *E. ciliata*. MCM is mostly wild, and JXR is the cultivated product of MCM, which was often confused with *Elsholtzia splendens* Nakai ex F. Maek. before [1]. However, Ganpei Zhu believes that JXR has obvious plant morphological differences from MCM. The plant height of JXR can reach 25–66 cm. The stem has gray, white curly pubescence. The leaf blade is broadly lanceolate to lanceolate, and the leaf margin is obviously serrate. The bracts are obovate and ovate. Calyx lobes are triangular lanceolate in shape. There is a hair ring at the base of the crown tube. Nutlets are yellowish brown and nearly round, with lightly carved surface, reticulate and flattened inside. MCM plants are shorter. Stem inversely pilose. Leaf blade linear to linear lanceolate, leaf margin serrate, inconspicuous. Bracts ovate orbicular. Calyx lobes subulate. There is no hairy ring at the base of the crown. Nutlets are nearly spherical, brown, with deep carving on the surface, and uneven in the mesh [2]. Therefore, JXR should be listed as an independent variety [3].

*E. ciliata* is a herbaceous plant distributed in Russia (Siberia), Mongolia, Korea, Japan, India, the Indochina peninsula, and China, while in Europe and North America it was also introduced and cultivated. In China, it is produced almost all over the country, except Xinjiang and Qinghai. It has low requirements for growth environment, a short growth cycle, flowering period from July to October, and harvest in summer and autumn [4,5,6]. 

Traditional Chinese medicine theory believes that *E. ciliata* has a spicy flavour and a lukewarm nature. It also has the effect of inducing diaphoresis and relieving superficies, removing dampness for regulating the stomach and inducing diuresis for removing edema. The following is a review of chemical compositions and pharmacological activities.

## 2. Chemical Constituents

A total of **352** compounds have been identified from *E. ciliata*. Among the chemical components of *E. ciliata*, flavonoids and terpenoids are the main components, which make *E. ciliata* have more obvious antimicrobial, anti-inflammatory, and antioxidant effects. Terpenoids such as 3-carene and some aromatic compounds such as carvacrol exhibit antimicrobial activity. Some polysaccharides can inhibit the proliferation of tumor cells, and show positive effects in immunoregulation.

Compounds **1**–**48** are flavonoids, **49**–**77** are phenylpropanoids, **78**–**193** are terpenoids, **194**–**202** are alkaloids compounds, and **203**–**352** are other compounds. Compounds **1**–**352** are listed in Table 1 and the structures **1**–**352** are listed in Figure 1. 

## 3. Pharmacological Activities

In the traditional application of Chinese medicine, *E. ciliata* is mainly used for the treatment of summer cold, cold aversion and fever, headache without sweat, abdominal pain, vomiting and diarrhea, edema, and poor urination. Modern pharmacological studies show that *E. ciliata* has antioxidant, anti-inflammatory, antimicrobial, insecticidal, antiviral, hypolipidemic, hypoglycemic, analgesic, antiarrhythmic, antitumor, antiacetylcholinesterase, and immunoregulator activities. 

### 3.1. Antioxidant Activity

Oxidative stress refers to a state of imbalance between oxidation and antioxidant effects in vivo. It is a negative effect caused by free radicals in the body and is considered to be an important factor in aging and disease. It was reported that the essential oil of *E. ciliata* could increase catalase (CAT) activity in brain of mice by 26.94%, which may be related to the decomposition of hydrogen peroxide by CAT to reduce oxidative stress [7]. There is a phenolic substance osmundacetone in *E. ciliata* ethanol extract. In DPPH experiment, the IC_50_ value of osmundacetone was 7.88 ± 0.02 µM, indicating a certain antioxidant capacity. The inhibitory effect of osmundacetone on glutamate-induced oxidative stress in HT22 cells was studied by reactive oxygen species (ROS) method. The results showed that osmundacetone significantly reduced the accumulation of ROS and could be used as a potential antioxidant [8]. By studying the effect of *E. ciliata* methanol extract on J774A.1 murine macrophage, the evaluation of antioxidant activity showed that all the tested compounds had significant effects on ROS release under oxidative stress at the highest concentration (10 M), especially luteolin-7-*O*-*β*-D-glucopyranoside, luteolin, and 5,6,4’-trihydroxy-7,3’-dimethoxyflavone [9]. 

Various scholars studied different polarity extracts of *E. ciliata*. According to the free radical scavenging experiment of Huynh Xuan Phong, the result showed that *E. ciliata* extract had certain scavenging ability against 2,2-diphenyl-1-picrylhydrazyl (DPPH) and 2,2’-azino-bis (3-ethylbenzothiazoline -6-sulfonic acid) (ABTS), with IC50 values of 495.80 ± 17.16 and 73. 59 ± 3.18 mg/mL. [10]. In DPPH experiment, the EC_50_ values of dichloromethane extract, crude ethanol extract and n-hexane extract were 0.041 µg/µg, 0.15 µg/µg and 0.46 µg/µg, respectively, showing strong antioxidant activity. Such antioxidant capacity may be related to non-polar flavonoids and phenols contained in *E. ciliata*, among which the total phenol content of dichloromethane is 96.68 ± 0.0010 µg GAEs/mg Extract, and the total flavonoid content is 71.5 ± 0.0089 µg QEs/mg Extract. Therefore, it has the strongest antioxidant capacity [11]. Jing-en Li et al. extracted JXR ethanol extract with petroleum ether, ethyl acetate and water-saturated n-butanol, respectively, and studied the antioxidant activities of the three parts and water phase. The results indicated that ethyl acetate showed good antioxidant activities in ferric reducing antioxidant power (FRAP), DPPH, and *β*-carotene assay, which may be related to the higher flavonoid content in this extract [12]. The antioxidant ability of mytilus polysaccharide-I (MP-I) contained in JXR water extract was concentration-dependent. When the concentration was 16 mg/mL, the chelation rate of MP-I and Fe^2+^ was 87.80%. When the concentration was 20 mg/mL, the scavenging rate of DPPH free radical was 81.32%. The scavenging rate of hydroxyl radical was 81.94% [13]. The DPPH test IC_50_ of MCM essential oil and methanol extract were 1230.4 ± 12.5 and 1482.5 ± 10.9 μg/mL, respectively. Reducing power test EC_50_ were 105.1 ± 0.9 and 313.5 ± 2.5 μg/mL, respectively. *β*-Carotene bleaching assay EC_50_ were 588.2 ± 4.2 and 789.4 ± 1.3 μg/ml, respectively. The total phenolic content of the essential oil was about 1.7 times that of methanol extract, which further verified the stronger antioxidant capacity of the essential oil [14]. 

Different parts of *E. ciliata* have different antioxidant capacity. Lauryna Pudziuvelyte et al. used DPPH, ABTS, FRAP, and cupric ion reducing antioxidant capacity (CUPRAC) to evaluate the antioxidant activity of different parts of *E. ciliata* DPPH and ABTS results showed that total phenolics content (TPC) and total flavonoids content (TFC) amounts of ethanol extracts from *E. ciliata* flower, leaf and whole plant were the highest and had the strongest antioxidant activity. The results of FRAP and CUPRAC test showed that the ethanol extract of *E. ciliata* flower had the highest antioxidant activity. Among different parts of the ethanol extract, the content of quercetin glycosides, phenolic acids, TPC, and TFC in stem extract was the lowest, and the antioxidant activity was the lowest [4]. The ethyl acetate fraction of *E. ciliata* was purified by macroporous resin with 80% ethanol to obtain fraction E. In the DPPH experiment, the EC_50_ value of fraction E was 0.09 mg/mL, which showed the strongest antioxidant and free radical scavenging ability. The EC_50_ value of fraction E was higher than positive control butylated hydroxytoluene (0.45), butylated hydroxyanisole (0.21), and vitamin C (0.41). Hence, it can be seen that *E. ciliata* has the potential to prevent cardiovascular diseases, cancer, and other diseases caused by excess free radicals [15]. 

### 3.2. Anti-Inflammatory Activity

Compounds pedalin, luteolin-7-*O*-*β*-D-glucopyranoside, 5-hydroxy-6,7-dimethoxyflavone, and *α*-linolenic acid in the essential oil of *E. ciliata* were investigated under a lipopolysaccharide (LPS)-induced inflammatory reaction. It can inhibit ROS release, but its mechanism deserves further study [9]. LPS-induced inflammation was evaluated by the number of inflammatory mediators, i.e., tumor necrosis factor-*α* (TNF-*α*), interleukin (IL)-6, and prostaglandin E2 (PGE2). *E. ciliata* ethanol extract could significantly inhibit the secretion of inflammatory mediators, where TNF-*α* and IL-6 factors could be effectively inhibited in the stem and flower part, and the PGE2 pathway could be inhibited in the leaf part [4]. The effect of *E. ciliata* on inflammation can be further verified by studying pyretic rats caused by LPS and mononuclear macrophage RAW264.7 induced by LPS. *E. ciliata* essential oil and water decoction can reduce the contents of PGE2, TNF-*α* and other inflammatory factors to different degrees, and can reduce the content of nitric oxide (NO) in serum [16]. Excessive NO can induce the production of pro-inflammatory factors, such as TGF-*α* and IL-1*β*, and aggravate the inflammatory response [17]. JXR alleviates dextran sulfate sodium induced intestinal knot inflammation in mice by affecting the release of NO, PGE2 and other inflammatory mediators and cytokines [18]. Carvacrol in MCM can inhibit the expression of pro-inflammatory cytokines interferon-*γ* (IFN-*γ*), IL-6, and IL-17 and up-regulate the expression of anti-inflammatory factors TGF-*β*, IL-4, and IL-10, thus reducing the level of inflammatory factors, reducing the damage to cells, and achieving anti-inflammatory effects [19]. 

In the formalin-induced licking response test, the licking time of *E. ciliata* crude ethanol extract and dichloromethane extract is shortened at the late phase under 100 mg/kg dose, and the licking time of n-hexane extract is shortened at the early phase under 100 mg/kg dose, which may be related to its anti-inflammatory effect [11]. Water extract of *E. ciliata* has anti-allergic inflammatory activity and may be related to the inhibition of calcium, P38 mitogen-activated protein kinase, and nuclear factor-*κ*B expression in the human mast cell line [20]. 

### 3.3. Antimicrobial Activity

Different polar extracts of *E. ciliata* demonstrated significant differences in inhibition ability with regard to microorganism. The results showed that dichloromethane fraction had the strongest inhibitory activity on *Candida albicans* with minimum inhibitory concentration (MIC) of 62.5 µg/mL, while n-hexane fraction had the strongest inhibitory effect on *Escherichia coli* with MIC of 250 µg/mL [11]. The ethyl acetate extract of JXR had strong inhibitory effect on *Rhizopus oryzae*, with the inhibition zone diameter of 13.7 ± 2.7 mm, MIC of 5 mg/mL and minimum bactericidal concentration (MBC) of 5 mg/mL [12]. The MIC of JXR petroleum ether extract, n-butanol extract and ethanol extract against *Escherichia coli*, *Staphylococcus aureus* and *Bacillus subtilis* were 31.25 μg/mL, and the MIC of ethyl acetate extract was 15.60 μg/mL [21]. The carbon dioxide extract of *E. ciliata* demonstrated a certain inhibitory effect on *Staphylococcus aureus*, *Salmonella paratyphoid,* and other microorganisms. When the concentration of the extract was 0.10 g/mL, the inhibitory effect on *Staphylococcus aureus* was the most obvious, and the diameter of the inhibition zone is 19.7 ± 0.1 mm [22].

According to existing research reports, *E. ciliata* is rich in essential oil, which contains abundant antibacterial ingredients and can inhibit a variety of microorganisms, so it has research significance and value. The main antibacterial active components of the essential oil of *E. ciliata* are thymol, carvacrol, and p-Cymene, which have inhibitory effect on *Staphylococcus aureus*, *Methicillin-resistant Staphylococcus aureus* and *Escherichia coli*. MIC were 0.39 mg/mL, 3.12 mg/mL and 1.56 mg/mL, and the diameters of inhibition zone were 21.9 ± 0.1230, 18.2 ± 0.0560, and 16.7 ± 0.0115 nm, respectively [23]. The essential oil in *E. ciliata* flowers, stems, and leaves had inhibitory effects on *Escherichia coli*, *Staphylococcus aureus*, *Salmonella typhi*, *Klebsiella pneumoniae,* and *Pseudomonas aeruginosa.* Both of them had the strongest inhibitory effect on *Staphylococcus aureus* with the inhibitory zone diameter of 12.2 ± 0.4 and 11.2 ± 0.1 mm, respectively [24]. Other relevant findings suggest that JXR essential oil may affect the formation of *Staphylococcus aureus* biofilm, so as to achieve bacteriostatic effect on its growth. The MIC of JXR essential oil to *Staphylococcus aureus* was 0.250 mg/mL. When the concentration was 4MIC, the inhibition rate of essential oil to *Staphylococcus aureus* biofilm formation could reach 91.3%, and the biofilm clearance rate was 78.5%. The MIC of carvacrol, thymol, and carvacrol acetate against *Staphylococcus aureus* were 0.122, 0.245, and 0.195 mg/mL, respectively, which were the effective antibacterial components of essential oil. Carvacrol, carvacryl acetate, *α*-cardene, and 3-carene had strong inhibitory effects on the formation of *Staphylococcus aureus* biofilm, and the inhibition rates were more than 80% at 1/4 MIC (0.0305, 1.4580, 0.1267 and 2.5975 mg/mL, respectively) [25,26]. In another study, Li Cao et al. studied the inhibitory effect of MCM essential oil on 17 kinds of microorganisms, among which, It significantly inhibited *Chaetomium globosum, Aspergillus fumigatus* and *Candida rugosa*. The antibacterial zone diameters were 16.3 ± 0.58, 15.0 ± 1.00, 16.0 ± 0.00 mm, and MIC were 31.3, 62.5, 62.5 μg/mL, respectively [14]. It also has obvious inhibitory effect on *Bacillus subtilis* and *Salmonella enteritidis*, which might be related to the terpenes contained, but this opinion remains to be verified [27]. Thymol and carvacrol are the main antibacterial components of MCM. Caryophyllene oxide can be used in the treatment of dermatomycosis, especially in the short-term treatment of mycosis ungualis [28]. The bactericidal mechanism of essential oil may be due to the fact that active components such as carvacrol can damage cell membranes and alter their permeability [29].

The extract of MCM had a significant inhibitory effect on the spore germination of *Aspergillus flavus* and could significantly change the morphology of *Aspergillus flavus* mycelia, podocytes, and sporophytes, with a MIC of 0.15 mg/mL [30]. The germination rate of *Penicillium digitorum* treated with carvacrol significantly decreased, the mechanism may be that carvacrol can change the surface morphology of mycelia, and the cavity rate of mycelia increased with the increase of carvacrol concentration. The permeability of the cell membrane of bacteria increases, causing an electrolyte imbalance in bacteria. As a result, the sugar content and nutrients in bacteria are reduced, so as to achieve bacteriostasis. The MIC and MBC of carvacrol against *Penicillium digitorum* were 0.125 and 0.25 mg/mL, respectively [31]. 

### 3.4. Insecticidal Activity

Some studies have shown that *E. ciliata* has an insecticidal effect. The repellency rate of *E. ciliata* essential oil to *Blattella germanica* was 64.50%, no significant difference from positive control diethyltoluamide (DEET) (*p* < 0.05). RD_50_ of *E. ciliata* essential oil was 218.634 µg/cm^2^, which was better than DEET (650.403 µg/cm^2^) [32]. Contact toxicity IC_50_ of *E. ciliata* essential oil to *Liposcelis bostrychophila* was 145.5 μg/cm^2^, and fumigation toxicity IC_50_ was 475.2 mg/L. (R)-carvone. Dehydroelsholtzia ketone and elsholtzia ketone are the active components of *E. ciliata* essential oil against *Liposcelis bostrychophila*. The IC_50_ of contact toxicity were 57.0, 151.5, and 194.1 μg/cm^2^, and those of fumigantion toxicity were 417.4, 658.2, and 547.3 mg/L, respectively [6]. Carvone and limonene are the two main components in *E. ciliata* essential oil. The ability of *E. ciliata* essential oil, carvone, and limonene against *Tribolium castaneum* larvae and adults was evaluated by a contact toxicity test and fumigation assay. Contact toxicity test showed that the LD_50_ of *E. ciliata* essential oil, carvone, and limonene to *Tribolium castaneum* adults were 7.79, 5.08, and 38.57 mg/Adult, respectively, and 24.87, 33.03, and 49.68 mg/Larva to *Tribolium castaneum* larvae. The results of fumigation toxicity test showed that LC_50_ of *Tribolium castaneum* adults were 11.61, 4.34, and 5.52 mg/L Air, respectively, and LC_50_ of *Tribolium Castaneum* larvae were 8.73, 28.71, and 20.64 mg/L Air, respectively [5]. Thymol, carvacrol, and *β* -thymol contained in JXR essential oil had significant fumigation toxicity against *Mythimna Separate, Myzus Persicae, Sitophilus Zeamais, Musca domestica,* and *Tetranychus cinnabarinus*, among which *β*-thymol has the strongest activity. The IC_50_ values for the five pests were 10.56 (9.26–12.73). 14.13 (11.84–16.59), 88.22 (78.53–99.18), 10.05 (8.63–11.46), and 7.53 (6.53–8.79) μL/L air, respectively [33]. Determined by the immersion method, the LC_50_ of MCM essential oil against *Aedes albopictus* larvae and pupae at four instars were 78.820 and 122.656 μg/mL, respectively. The chemotaxis activity of MCM essential oil was evaluated by the method of effective time of human local skin coating. When the dose was 1.5 mg/cm^2^, the complete protection time of *Aedes albopictus* was 2.330 ± 0.167 h [34]. From this point of view, *E. ciliata* essential oil has the development potential as a natural anti-insect agent. It provides a basis for the development and utilization of pesticide dosage forms.

*Leishmania mexicana* can cause cutaneous leishmaniasis. *E. ciliata* essential oil had anti-leishmania activity with IC_50_ of 8.49 ± 0.32 nL/mL. *Leishmania mexicana mexicana* was treated with a survival rate of 0.38 ± 0.00 %. Selectivity indices were 5.58 and 1.56 for mammalian cell WI38 and J774, respectively. This provides a reference for the treatment of cutaneous leishmaniasis [35]. *E. ciliata* water extract has an obvious anti-*trichomonas vaginalis* effect, i.e., can destroy the insect body structure, to achieve the purpose of killing insects. The results of in vitro experiments showed that the lowest effective concentration of *E. ciliata* water extract was 62.5 mg/mL, and the lowest effective time was 12 h. When the concentration was 250 mg/mL, all *Trichomonas vaginalis* could be killed for 4 h. This experiment provides a new idea for the clinical treatment of vaginal trichomoniasis [36]. 

### 3.5. Antiviral Activity

T helper 17 (Th17) cells play an important role in maintaining adaptive immune balance, and an excess of Th17 cells can cause inflammation. Carvacrol plays an anti-influenza virus role by reducing the proportion of Th17 cells significantly increased by influenza virus A infection. It can be used as a potential antiviral drug and can also be used to control inflammation caused by influenza virus A infection [19]. Mice with viral pneumonia modeled by A/PR/8/34 (H1N1) virus were treated with low, medium and high dose of MCM total flavonoids. Lung index of the three dose groups were 12.81 ± 3.80, 11.65 ± 2.58, 11.45 ± 2.40 mg/g, respectively, compared with the infection group 16.05 ± 3.87 mg/g, the inhibition rates were 20.18%, 27.41%, 28.66%, respectively [37]. *E. ciliata* ethanol extract has an inhibitory effect on the proliferation of avian infectious bronchitis virus, which may be related to the increased expression of three antiviral genes suppressor of cytokine signaling 3 (SOCS3), 2′-5′-oligoadenylate synthetase-like (OASL), and signal transducer and activator of transcription 1 (STAT1) in H1299 cells treated with extract, and this inhibitory effect shows a certain concentration dependence. In addition, the extract had no cytotoxicity when the concentration was less than 0.3 g/mL [38]. Above experiments provide new possibilities for the treatment of inflammation caused by the virus.

A/WSN/33/2009 (H1N1) virus was used to infect Madin-Darby canine kidney cells to explore the antiviral activity of phenolic acids from MCM in vitro. The survival rate of the cells treated with the compound 3-(3,4-dihydroxyphenyl) acrylic acid 1-(3,4-dihydroxyphenyl)-2-methoxycarbonylethyl and methyl lithospermate were higher than 80%, and the inhibition rate of virus at 100 μmol/L were 89.28% and 98.61%, respectively [39]. In another study, the lung index of low, medium, and high dose of MCM water extract on mice infected by A/PR8 influenza virus was 1.21 ± 0.22%, 1.12 ± 0.17%, and 0.94 ± 0.21%, respectively. Compared to the virus-infected group 1.80 ± 0.29 %, the inhibition rates were 32.78%, 37.78% and 47.78%, respectively. The extracts of the three groups can increase the amounts of IL-2 and IFN-*γ* in serum of mice, and promote the antiviral ability of the body indirectly or directly [40]. Fluoranthene is a compound with antiviral activity extracted from *E. ciliata*. It has a certain inhibitory effect on two enveloped viruses, sindbis virus, and murine cytomegalovirus, with the lowest effective concentrations of 0.01 and 1.0 μg/mL, respectively. However, its biological effects are complex, and its clinical safety and effectiveness need further research [41].

### 3.6. Hypolipidemic Activity

The hypolipidemic activity of *E. ciliata* ethanol extract was evaluated by determining the effects on the contents of triglyceride and total cholesterol in serum of mice in vivo and the proliferation of 3T3-L1 preadipocytes in vitro. The results showed that the levels of triglyceride and total cholesterol in serum of mice treated with the extract were decreased, and the differentiation and accumulation of 3T3-L1 preadipocytes were also effectively inhibited. The levels of genes associated with adipogenesis, such as peroxisome proliferator activated receptor *γ* (PPAR*γ*), fatty acid synthase (FAS), and adipocyte fatty acid-binding protein 2 (aP2) were also significantly reduced. In addition, serum leptin content in *E. ciliata* ethanol extract treatment group was lower than that in obese mice, which may be due to the reduction of fat content. By this token, the action mechanism of *E. ciliata* lowering blood lipids may be to inhibit the expression of genes related to fat cell formation. However, the specific mechanism needs further study [42]. 

### 3.7. Antitumor Activity

Pudziuvelyte, L. et al. extracted essential oil from *E. ciliata* fresh herbs, lyophilized herbs, and dried herbs, respectively. In in vitro experiments, three kinds of essential oil presented significant inhibition of proliferation effect on the human glioblastoma (U87), pancreatic cancer (PANC-1), and triple negative breast cancer (MDA-MB231) cells, with EC_50_ values ranging from 0.017% to 0.021%. However, *E. ciliata* ethanol extract did not show cytotoxicity in this experiment [43]. The antitumor activity of origin processing integration technology and traditional cutting processing technology of *E. ciliata* was evaluated by measuring the effect of the decoction and essential oil on the average optical density of TNF-*α* in rat lung tissue. Average optical densities of water decocted solution and essential oil of traditional cutting *E. ciliata* were 0.530 ± 0.071 and 0.412 ± 0.038, respectively, and those of integration processing technology of origin were 0.459 ± 0.051 and 0.459 ± 0.051, respectively. Compared with the blank group (0.299 ± 0.028), there were varying degrees of increase [44]. In vitro experiments of JXR pectin polysaccharide (MP-A40) showed that the proliferation of human leukemic cell line K562 was affected by MP-A40. When the concentration of MP-A40 was 500μg/mL, the inhibition rate was 31.32% [45]. 

### 3.8. Immunoregulatory Activity

Macrophages can regulate apoptosis by producing NO and other effecting molecules. Macrophage RAW 264.7 cells treated by JXR pectin polysaccharide (MP-A40) showed an obvious increase in NO production. Moreover, it’s concentration-dependent. When the concentration of MP-A40 was as low as 10 μg/mL, NO production was still 15 times that of negative control [45]. Mice treated with cyclophosphamide had elevated levels of free radicals, increasing aggression towards immune organs, and decreased thymus and spleen indices. Polysaccharide MP can scavenge free radicals and promote the proliferation of ConA-induced T cells and LPS-induced B cells. To a certain extent, the immunosuppression induced by cyclophosphamide can be alleviated [13,46]. However, the potential immunomodulatory mechanism of polysaccharide remains to be further studied.

### 3.9. Others

Different polar ethanol extracts of JXR had different degrees of inhibition on *α*-glucosidase activity. Therefore, it has certain hypoglycemic activity. When the polar ethanol extract concentration was 4.0 mg/mL, the inhibition rate of petroleum ether extract was 93.8%, IC_50_ was 0.339 mg/mL, and the inhibition rate of ethyl acetate extract was 92.8%, IC_50_ was 0.454 mg/mL. The essential oil prepared by steam distillation, petroleum ether cold extraction, and petroleum ether reflux extraction also showed significant inhibition of *α*-glucosidase at the concentration of 0.25 mg/mL, and the inhibition rates were more than 90% [47]. 

The results of the formalin-induced Licking test showed that *E. ciliata* crude ethanol extract has analgesic effect on the early stage of reaction (0–5 min) [11]. 

THE Langendorff perfused isolated rabbit heart model was used. When *E. ciliata* essential oil was added into perfusate, QRS interval was increased, QT interval was shortened, AND action potentials upstroke amplitude was decreased, and activation time was prolonged when the concentration of *E. ciliata* essential oil was increased in the range of 0.01–0.1 μL/mL, and showed concentration dependence. This may be due to the fact that sodium channel block can increase the threshold of action potential generation, prolong the effective refractory period, and inhibit the zero-phase depolarization of late depolarization. The reduction of action potential duration can reduce the occurrence of early depolarization. This experiment provides theoretical basis for *E. ciliata* in the treatment of arrhythmia [48].

7-*O*-(6-*O*-acetyl)-*β*-D-glucopyranosyl-(1→2)[(4-oacetyl)-*α*-L-rhamnopyranosyl-(1→6)]-*β*-D-glucopyranoside in methanol extract of *E. ciliata* was hydrolyzed to obtain acacetin. The IC_50_ of acacetin against acetylcholinesterase was 50.33 ± 0.87 μg/mL, which showed a significant inhibitory effect on acetylcholinesterase activity, which may hold promise for Alzheimer’s disease treatment [49].
molecules-27-06411-t001_Table 1Table 1Compounds [5,6,9,11,18,22,24,26,27,28,33,34,39,43,49,50,51,52,53,54,55,56,57,58,59,60,61,62,63,64,65,66,67,68,69].No.Compound NameFormula*E. ciliata*JXRMCMReferencesFlavonoids1vitexinC_21_H_20_O_10_+--[11]2pedalin C_22_H_22_O_12_+--[11]3luteolin-7-*O*-*β*-D-glucopyranoside C_20_H_20_O_11_+--[11]4apigenin-5-*O*-*β*-D-glucopyranosideC_20_H_20_O_10_+--[11]5apigenin-7-*O*-*β*-D-glucopyranoside C_20_H_20_O_10_+--[11]6chrysoeriol-7-*O*-*β*-D-glucopyranosideC_21_H_22_O_11_+--[11]77,3′-dimethoxyluteolin-6-*O*-*β*-D-glucopyranosideC_21_H_24_O_12_+--[11]8luteolinC_15_H_10_O_6_++-[11,18]95,6,4′-trihydroxy-7,3′-dimethoxyflavoneC_17_H_14_O_7_+--[9]105-hydroxy-6,7-dimethoxyflavoneC_17_H_14_O_5_+--[9]115-hydroxy-7,8-dimethoxyflavoneC_17_H_14_O_5_+--[9]12negleteinC_16_H_12_O_5_--+[50]13acacetin-7-*O*-[*β*-D-glucopyranosyl(1″″→2″)-4‴-*O*-acetyl-*α*-L-rhamnopyranosyl(1‴→6″)]-*β*-D-glucopyranosideC_36_H_44_O_20_+--[51]14acacetin-7-*O*-[6″″-*O*-acetyl-*β-*D-glucopyranosyl(1″″→2″)-*α*-L-rhamnopyranosyl(1‴→6″)]-*β-*D-glucopyranosideC_36_H_44_O_20_+--[51]15acacetin-7-*O*-[6″″-*O*-acetyl-*β*-D-glucopyranosyl(1″″→2″)-3‴-*O*-acetyl-*α*-L-rhamnopyranosyl(1‴→6″)]-*β*-D-glucopyranosideC_38_H_46_O_21_+--[51]16acacetin-7-*O*-[6″″-*O*-acetyl-*β*-D-glucopyranosyl(1″″→2″)-4‴-*O*-acetyl-*α*-L-rhamnopyranosyl(1‴→6″)]-*β*-D-glucopyranosideC_38_H_46_O_21_+--[51]17acacetin-7-*O*-[3″″,6″″-di-Oacetyl-*β*-D-glucopyranosyl(1″″→2″)-4‴-*O*-acetyl-*α*-L-rhamnopyranosyl(1‴→6″)]-*β*-D-glucopyranosideC_40_H_45_O_25_+--[51]187-*O*-*β*-D-glucopyranosyl-(1 → 2)[*α*-L-rhamnopyranosyl(1 → 6)]-*β*-D-glucopyranosideC_34_H_42_O_19_+--[49]19linarinC_28_H_32_O_14_+--[51]20acacetin-7-*O*-[4‴-*O*-acetyl-*α*-L-rhamnopyranosyl(1‴→6″)]-*β*-D-glucopyranosideC_30_H_34_O_15_+--[51]21apigeninC_15_H_10_O_5_+-+[50,51]22apigetrinC_21_H_20_O_10_+--[51]23diosmetinC_16_H_12_O_6_+--[51]245-hydroxy-6,7-dimethoxyflavoneC_17_H_14_O_5_+--[51]255-hydroxy-7,8-dimethoxyflavoneC_17_H_14_O_5_+--[51]266-hydroxy-5,7,8-trimethoxyflavoneC_18_H_16_O_6_+--[51]27butinC_16_H_14_O_4_+--[51]28isookaninC_16_H_14_O_5_+--[51]29sulfuretinC_15_H_10_O_5_+--[51]303,2′,4′-trihy-droxy-4-methoxychalconeC_16_H_14_O_5_+--[51]313,2′,4′-trihy4′-*O*-*β*-D-glucopyranosyl-3,2′-dihydroxy-4-methoxychalconeC_21_H_24_O_10_+--[51]32okanin-4-methoxy-4′-*O*-*β*-D-glucopyranosideC_21_H_24_O_11_+--[51]33neoisoliquiritinC_20_H_22_O_9_+--[51]34kumatakeninC_17_H_14_O_6_+--[52]35isoorientin-2′′-*O*-rhamnosideC_27_H_30_O_15_-+-[18]36isovitexin-2′′-*O*-rhamnosideC_27_H_30_O_14_-+-[18]37quercetin-3-*O*-rutinosideC_27_H_30_O_16_-+-[18]38quercetin-3-*O*-glucosideC_21_H_20_O_12_-+-[18]39orientin-2′′-*O*-rhamnosideC_27_H_30_O_15_-+-[18]40vitexin-2′′-*O*-rhamnosideC_27_H_30_O_14_-+-[18]41orientinC_21_H_20_O_11_-+-[18]42isoorientinC_21_H_20_O_11_-+-[18]43swertisinC_22_H_22_O_10_-+-[18]44peonidin-3-*O*-glucosideC_22_H_22_O_11_-+-[18]45chrysoeriolC_16_H_12_O_6_-+-[50]46quercetinC_15_H_10_O_7_-+-[50]47kaempferolC_15_H_10_O_6_-+-[53]48catechin C_15_H_14_O_6_+--[54]Phenylpropanoids49caffeic acidC_9_H_8_O_4_++-[9,18]50(E)-p-coumaric acidC_9_H_8_O_3_+--[9]51osmundacetoneC_10_H_10_O_3_+--[9]524-(E)-caffeoyl-L-threonic acidC_17_H_16_O_12_+--[9]534-*O*-(E)-p-coumaroyl-L-threonic acidC_13_H_14_O_7_+--[9]54(7E,9E)-3-hydroxyavenalumic acid-3-*O*-[6′-*O*-(E)-caffeoyl]-*β*-D-glucopyranosideC_26_H_26_O_12_+--[51]55gnaphaliin CC_25_H_24_O_11_+--[51]563,5-di-*O*-caffeoylquinic acidC_25_H_24_O_12_+--[51]57everlastoside LC_25_H_26_O_11_+--[51]581-(3′,4′-dihydroxycinnamoyl)cyclopentane2,3-diolC_14_H_16_O_6_+--[51]59ethyl caffeateC_11_H_12_O_4_+--[51]60(Z)-p-coumaric acidC_9_H_8_O_3_+--[51]61p-hydroxybenzaldehydeC_7_H_6_O_2_+--[51]62p-hydroxybenzoic acidC_7_H_6_O_3_++-[18,51]633,4-dihydroxybenzoic acidC_7_H_6_O_4_+--[51]64vanillic acidC_8_H_8_O_4_+--[51]65rosmarinic acidC_18_H_16_O_8_++-[18,51]66estra-1,3,5(10)-trien-17-*β*-olC_18_H_24_O+--[55]675-caffeoylquinic acidC_16_H_17_O_9_-+-[18]684-caffeoylquinic acidC_16_H_17_O_9_-+-[18]69DanshensuC_9_H_10_O_5_-+-[18]70(+)lyoniresinolC_24_H_38_O_8_-+-[53]71(-)-5-methoxyisolariciresinolC_22_H_30_O_7_-+-[53]72pinoresinolC_21_H_26_O_6_-+-[53]73isoeucommin AC_27_H_34_O_12_-+-[53]74episyringaresinol-4-*O*-*β*-D-glucopyranosideC_28_H_36_O_13_-+-[53]75methyl-3-(3′,4′dihydroxyphenyl) lactate C_10_H_12_O_5_-+-[56]76(s)-pencedanol-7-*O*-*β*-D-glucopyranoside C_20_H_26_O_10_-+-[56]77stearyl ferulate C_28_H_46_O_4_+--[54]Terpenoids(1) Monoterpene 78*α*-PineneC_10_H_16_+++[5,26,57]79*β*-PineneC_10_H_16_+-+[5,57]80myrceneC_10_H_16_+-+[5,57]81*β*-PhellandreneC_10_H_16_+--[5]82limoneneC_10_H_16_+-+[5,57]83*β*-OcimeneC_10_H_16_+-+[5,57]84campheneC_10_H_16_+-+[57,58]85isoterpinolene C_10_H_16_+--[58]86*γ*-terpinene C_10_H_16_++-[26,58]874-carene C_10_H_16_+--[6]88sabinene C_10_H_16_+-+[57,59]89*α*-phellandrene C_10_H_16_-++[26,57]903-carene C_10_H_16_-++[26,57]91sylvestrene C_10_H_16_-+-[26]92terpinolene C_10_H_16_-++[26,57]93*α*-thujene C_10_H_16_--+[57]94bicyclo [3.1.0]hex-2-ene, 4-methylene-1-(1-methylethyl)- C_10_H_14_--+[57]95bicyclo[4.2.0]oct-1-ene,7-exo-ethenyl- C_10_H_14_--+[57]96*α*-terpinene C_10_H_16_--+[57]972,6-dimethyl-1,3,5,7-octatetraene C_10_H_14_--+[57]98p-mentha-1(7),2-dieneC_10_H_16_+--[60]99cyclohexane, 1-methylene-4-(1-methylethenyl)- C_10_H_16_+--[61]100artemisia triene C_10_H_16_--+[28](2) Oxygenated monoterpene101linaloolC_10_H_18_O+-+[5,57]102elsholtzia ketoneC_10_H_14_O_2_+--[5]103carvoneC_10_H_14_O+--[5]104dehydroelsholtzia ketoneC_10_H_12_O_2_+--[5]105neodihydrocarveol C_10_H_18_O+--[58]106eucalyptol C_10_H_18_O++-[33,58]107menthone C_10_H_18_O+--[58]108linalool C_10_H_18_O+--[58]109camphor C_10_H_16_O+--[58]110eucarvone C_10_H_14_O+--[58]111perillene C_10_H_14_O+--[58]112jasmone C_10_H_14_O+--[58]113dehydroelsholtzia ketone C_10_H_12_O_2_+--[58]114eucalyptol C_10_H_18_O++-[33,58]115(−)-1R-8-hydroxy-p-menth-4-en-3-one C_10_H_16_O_2_+--[43]116fenchol C_10_H_18_O+--[6]117lavandulol C_10_H_18_O+--[6]118*α*-terpineol C_10_H_18_O+--[6]1191,6-dihydrocarveol C_10_H_18_O+--[6]120cis-carveol C_10_H_16_O+--[6]121terpinen-4-ol C_10_H_18_O++-[33,59]122nerol C_10_H_18_O+--[59]123neral C_10_H_16_O+--[59]124geraniol C_10_H_18_O+--[59]125borneol C_10_H_18_O-+-[33]126trans-4-thujanol C_10_H_18_O-+-[26]127sabinol C_10_H_16_O-+-[26]128thymoquinone C_10_H_12_O_2_--+[57]129umbellulon C_10_H_14_O--+[27]130furan,3-methyl-2-(3-methyl-2-buten-1-yl)-C_10_H_14_O+--[24]131verbenolC_10_H_16_O--+[28]132*β*-dehydro-elsholtzione C_10_H_12_O_2_+--[22]133rose furan epoxide C_10_H_14_O_2_+--[65]1341-octen-3-yl acetate C_10_H_18_O_2_+--[24]135neryl formate C_11_H_18_O_2_+--[59]136geranyl formate C_11_H_18_O_2_+--[59]137neryl acetate C_12_H_20_O_2_+--[59]138citronellal C_10_H_18_O+--[6]139isobornyl formate C_11_H_18_O_2_--+[57](3) Sesquiterpene140cubebeneC_15_H_24_+--[5]141*β*-bourboneneC_15_H_24_+--[5]142*β*-caryophylleneC_15_H_24_+++[5,26,57]143*α*-caryophylleneC_15_H_24_++-[5,33]144*α*-farneseneC_15_H_24_+++[5,26,57]145*β*-bourbonene C_15_H_24_+--[58]146*β*-gurjunene C_15_H_24_+--[58]147*π*-cubebene C_15_H_24_+--[58]148*α*-bergamotene C_15_H_28_++-[26,58]149humulene C_15_H_24_+--[58]150*π*-sesquiphellandrene C_15_H_24_+--[58]151germacrene D C_15_H_24_++-[26,58]152*π*-bisabolene C_15_H_24_+--[58]153*γ*-elemene C_15_H_24_+--[58]154(Z, E)-*α*-farnesene C_15_H_24_+--[58]155caryophyllene C_15_H_24_++-[33,58]156*π*-muurolene C_15_H_24_+--[58]157*π*-cadinene C_15_H_24_+--[58]158*α*-farnesene C_15_H_24_+--[43]159ledene C_15_H_24_+--[43]160*α*-cubebene C_15_H_24_+--[43]161*γ*-cadinene C_15_H_24_+--[43]162*δ*-cadinene C_15_H_24_+--[43]163*β*-elemene C_15_H_24_+--[6]164aromadendrene C_15_H_24_+--[6]165*β*-bisabolene C_15_H_24_+--[59]166trans-*α*-bergamotene C_15_H_24_-+-[33]167*γ*-muurolene C_15_H_24_-+-[26]168eudesma-3,7(11)-diene C_15_H_24_-+-[26]169longifolene C_15_H_24_--+[57]170trans-*α*-bergamotene C_15_H_24_--+[57]171trans-*β*-bergamotene C_15_H_24_--+[57]172(Z,E)-*α*-farnesene C_15_H_24_--+[57]173*β*-himachalene C_15_H_24_--+[57]174*γ*-cadinene C_15_H_24_--+[57]175guaiene C_15_H_24_--+[64]176*α*-zingiberene C_15_H_24_--+[64]177*α*-muurolene C_15_H_24_--+[64]178*β*-selinene C_15_H_24_+--[65](4) Oxygenated sesquiterpene179(-)-humulene epoxide IIC_15_H_24_O+--[5]180nerolidol C_15_H_26_O+--[58]181caryophyllene oxide C_15_H_24_O++-[33,58]182spathulenol C_15_H_24_O+--[6]183humulene-1,2-epoxide C_15_H_24_O-+-[26]184cedrol C_15_H_26_O--+[57]185cedrenol C_15_H_24_O--+[64]186levomenol C_15_H_26_O--+[64]187germacrone C_15_H_22_O+--[22](5) Oxygenated diterpene1882,3-dimethyl-5-(2,6,10trimethylundecyl) furan C_20_H_36_O+--[43]189phytol C_20_H_40_O--+[28](6) Triterpene190forrestin A C_30_H_42_O_11_-+-[18]191betulinic acid C_30_H_48_O_3_-+-[50]192oleanolic acid C_30_H_48_O_3_-+-[50]193ursolic acid C_30_H_48_O_3_-+-[50]Alkaloids194N-trans-feruloyloctopamineC_18_H_19_NO_5_+--[51]1955-methyl-furan-2-carboxylic acid (1H-[1,2,4]triazol3-yl) -amide C_8_H_8_N_4_O_2_+--[51]196furane-2-carboxaldehyde, 5 (nitrophenoxymethyl) C_12_H_9_NO_5_+--[51]197uridine C_9_H_12_N_2_O_6_-+-[18]198carbamult C_12_H_17_NO_2_-+-[26]199phenol,O-amino- C_6_H_7_NO+--[61]200adenosinC_10_H_13_N_5_O_4_-+-[67]201prunasin C_14_H_17_NO_6_-+-[68]202sambunigrin C_13_H_17_NO_7_-+-[68]Others(1) Aliphatic203pentacosane C_25_H_52_+--[55]204heptacosane C_27_H_56_+--[55]205octacosane C_28_H_58_+--[55]206tetratriacontane C_34_H_70_+--[55]207hexatriacontane C_36_H_74_+--[55]208n-dodecane C_11_H_24_--+[57]209n-tridecane C_13_H_28_--+[57]210n-tetradecane C_14_H_30_--+[57]211n-pentadecane C_15_H_32_--+[57]212n-hexadecane C_16_H_34_--+[57]213heptadecane C_17_H_36_-+-[63]214octadecane C_18_H_38_-+-[63]215tridecane,4-cyclohexyl C_19_H_38_-+-[63]216nonadecane C_19_H_40_-+-[63]217eicosane C_20_H_42_-+-[63]218heneicosane C_21_H_44_-+-[63]219docosane C_22_H_46_-+-[63]220tricosane C_23_H_48_-+-[63]221tetracosane C_24_H_50_-+-[63]2229-cyclohexyl eicosane C_26_H_52_-+-[63]223(R, R)-2,3-butanediol C_4_H_10_O_2_+--[58]2243-hexen-1-ol C_6_H_12_O+--[58]2251-hexanol C_6_H_14_O+--[58]2263-octen-1-ol C_8_H_16_O+--[58]2273-octanol C_8_H_18_O+--[58]2283,13-octadecadien-1-ol C_18_H_34_O-+-[63]2292-octene-1-ol C_8_H_16_O+--[66]230hexanal C_6_H_12_O+--[58]231heptana C_7_H_14_O+--[58]232octanal C_8_H_16_O+--[58]233nonanal C_9_H_18_O+--[58]234nonacosan-10-one C_29_H_58_O+--[55]2353-hexanone C_6_H_12_O+--[58]2363-heptanone C_7_H_14_O+--[58]2373-octanone C_8_H_16_O+--[58]238irisone C_13_H_20_O+--[24]2392-pentadecanone,6,10,14-trimethyl C_18_H_36_O-+-[63]240*α*-linolenic acid C_18_H_30_O_2_+--[9]241n-hexadecanoic acid C_16_H_32_O_2_+--[55]2429,12-octadecadienoic acid C_18_H_32_O_2_+--[55]2433-methylpentanoic acid C_6_H_12_O_2_+--[58]2442-methylbutanoic acid C_5_H_10_O_2_+--[58]245galactonic acid C_6_H_12_O_7_-+-[18]246malic acid C_4_H_6_O_5_-+-[18]247citric acid C_6_H_8_O_7_-+-[18]248quinic acid C_7_H_12_O_6_-+-[18]249succinic acid C_4_H_6_O_4_-+-[18]250azelaic acid C_9_H_16_O_4_-+-[18]2519-hydroxy-10,12-octadecadienoic acid C_18_H_32_O_3_-+-[18]252palmitic acidC_16_H_32_O_2_-+-[18]253oleic acid C_18_H_34_O_2_-+-[18]254tetradecanoic acid C_14_H_28_O_2_-+-[63]255linoleic acid C_18_H_32_O_2_+

[66]256hexadecanoic acid, ethyl ester C_18_H_36_O_2_+--[55]2579,12,15-octadecatrienoic acid methyl ester C_19_H_32_O_2_+--[55]258linoleic acid ethyl ester C_20_H_36_O_2_+--[55]2599,12,15-octadecatrienoic acidethyl ester C_20_H_34_O_2_+--[55]260octadecanoic acidethyl ester C_20_H_40_O_2_+--[55]261linolenic acide, 2-hydroxy-1(hydroxymethyl) ethyl esterC_21_H_36_O_4_+--[55]262octen-1-ol, acetate C_10_H_18_O_2_+--[58]2633-octanol, acetate C_10_H_20_O_2_+--[58]2642-propenoic acid, 2-methyl-, ethenyl ester C_6_H_8_O_2_+--[43]265diglycol laurate C_16_H_32_O_4_-+-[18]266butanoic acid, 2-methylpropyl ester C_8_H_16_O_2_--+[57]267propanoic acid, 2-methyl-,butyl ester C_8_H_16_O_2_--+[57]2682-propenoic acid, 2-methyl-,butyl ester C_8_H_14_O_2_--+[57]269butanoic acid, butyl ester C_8_H_16_O_2_--+[57]270propanoic acid, 2-methyl,2-methylpropyl ester C_8_H_16_O_2_--+[57]271heptadecanoic acid, ethylester C_19_H_38_O_2_-+-[63]272acetic acid, bornyl ester C_12_H_20_O_2_--+[34]273ethyl linoleate C_20_H_36_O_2_+--[22]274octenylacetate C_10_H_18_O_2_+--[65]2752,5-diethyltetrahydrofuran C_8_H_16_O+--[58]2761,8-cineole C_10_H_18_O+--[6]277dihydroartemisinin ethyl ether C_17_H_28_O_5_-+-[18]278pentane, 1-butoxy- C_9_H_20_O--+[57]279ethane, 1,1-dibutoxy- C_10_H_22_O_2_--+[57]2801,1-dibutoxy-isobutane C_12_H_26_O_2_--+[57]281butane, 1,1-dibutoxy- C_12_H_26_O_2_--+[57]2823-methyl-3-oxetanemethanol C_5_H_10_O_2_+--[43]283neryl-*β*-D-glucopyranoside C_16_H_28_O_6_+--[51]284furfural C_5_H_4_O_2_+--[58]2852-acetyl-5-methylfuran C_7_H_8_O_2_+--[58]286geranyl acetate C_12_H_20_O_2_+--[58]287octen-3-ol C_9_H_19_O+-+[6,57]288cis-jasmone C_11_H_16_+--[6]2892,6-octadien-1-ol, 3,7-dimethyl-C_11_H_20_--+[57]2903,5-dimethylcyclohex-1-ene-4carboxaldehyde C_9_H_14_O-+-[33]291(6S,9R)-roseoside C_19_H_30_O_8_-+-[67]2921-hexadecene C_16_H_32_-+-[63]2931-nonene C_9_H_18_--+[34]294ligustilide C_12_H_14_O_2_+--[22]295cyclohexene, 2-ethenyl-1,3,3-trimethyl C_11_H_18_+--[43]296daucosterol C_35_H_60_O_6_+--[54](2) Aromatic297thymol C_10_H_14_O+++[33,57,58]298carvacrol C_10_H_14_O-++[33,57]299p-cymene C_10_H_14_+++[6,26,57]300methyl chavicol C_10_H_12_O+--[59]301m-cymene C_10_H_14_--+[57]3023,4-diethylphenol C_10_H_14_O--+[57]303p-eugenol C_10_H_12_O_2_--+[57]304O-cymeneC_10_H_14_+--[60]305cis-anethol C_10_H_12_O-+-[62]3063,4,5-trimethoxytoluene C_10_H_14_O_3_-+-[62]307cis-asarone C_10_H_14_O_2_-+-[63]308cuminaldehyde C_10_H_12_O--+[64]309benzyl-*β*-D-glucopyranoside C_13_H_18_O_6_+--[51]310p-xylene C_8_H_10_+--[58]311benzaldehyde C_7_H_6_O+-+[57,58]312thymol acetate C_12_H_16_O_2_+++[33,57,58]313acetophenone C_8_H_8_O+--[58]314naphthalene C_10_H_8_O+--[43]315protocatechuic acid C_7_H_6_O_4_-+-[18]316sodium ferulate C_10_H_9_NaO_4_-+-[18]3174,5-dimethyl-2-ethylphenol C_10_H_14_O-+-[33]318thymyl acetate C_12_H_16_O_2_-+-[26]319carvacryl acetate C_12_H_16_O_2_-+-[26]3204-*O*-*β*-D- glucopyranosylbenzyl-4′-hydroxylbenzoateC_20_H_22_O_9_--+[39]3214-*O*-*β*-D-glucopyranosylbenzyl-3′-hydroxy-4′- methoxybenzoateC_21_H_24_O_10_--+[39]322amburoside AC_20_H_22_O_10_--+[39]3234-[[(2′,5′-Dihydroxybenzoyl)oxy]methyl]phenyl-*O*-*β*-D-glucopyranosideC_20_H_20_O_10_--+[39]324hyprhombin B methyl esterC_26_H_22_O_9_--+[39]325methyl lithospermateC_28_H_24_O_12_--+[39]326dimethyl lithospermateC_29_H_26_O_12_--+[39]327salvianolic acid C methyl esterC_27_H_23_O_10_--+[39]328sebestenoids CC_36_H_30_O_14_--+[39]329cucurbitoside DC_25_H_30_O_13_--+[39]3302,4-di-tert-butylphenolC_14_H_22_O--+[27]331benzyl benzoate C_14_H_12_O_2_+--[69]3324-isopropyl-3-methylphenol C_10_H_14_O+--[60]333benzenemethanol, 4-(1-methylethyl)*-*C_10_H_14_O+--[61]334methyl eugenolC_11_H_14_O_2_-+-[62]335elemicin C_12_H_16_O_3_-+-[62]336myristicin C_11_H_12_O_3_-+-[62]337methyl thymol etherC_11_H_16_O--+[28]338apiole C_12_H_14_O_4_--+[28]3394-hydroxy-2,6-dimethoxyphenyl-*β*-D-glucopyranoside C_14_H_20_O_9_-+-[67]3404-hydroxy-3,5-dimethoxyphenyl-*β*-D-glucopyranosideC_14_H_20_O_9_-+-[67]3413,4,5-dimethoxyphenyl-*β*-D-glucopyranoside C_15_H_22_O_9_-+-[67]3423-hydroxyestragole-*β*-D-glucopyranosides C_16_H_22_O_7_-+-[67]3434-acetoxy-3-methoxy acetophenone C_11_H_12_O_4_-+-[63]344benzyl-D-glucopyranoside C_13_H_18_O_6_-+-[63]3452′,4′-dihydroxy-3′methylacetophenone C_9_H_10_O_3_--+[34]346acetylthymol C_12_H_16_O_2_--+[64]347carvacrol acetate C_12_H_16_O_2_+--[66]348bergaptene C_12_H_8_O_4_+--[66]3492-butenylbenzene C_10_H_12_+--[22]350*α*-methyl-benzenepropanol C_10_H_14_O+--[22]351cuminic acid C_10_H_12_O+--[22]352p-cymen-8-ol C_11_H_16_O+--[6]“+” Found in plants. “-” Not found in plants.
Figure 1Chemical structures isolated from *E. ciliata*.
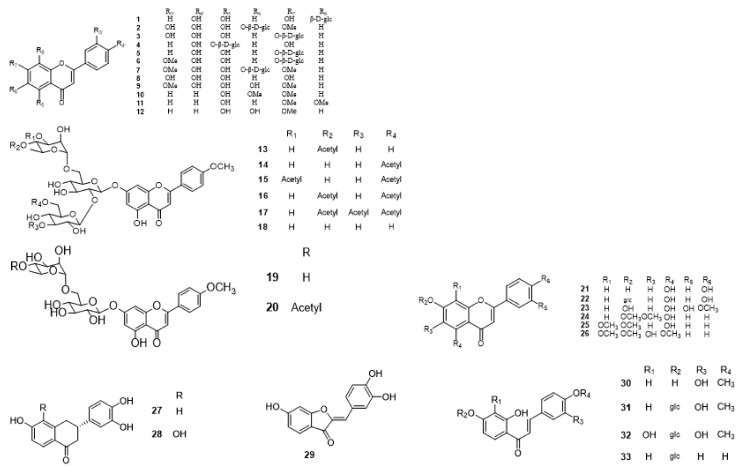

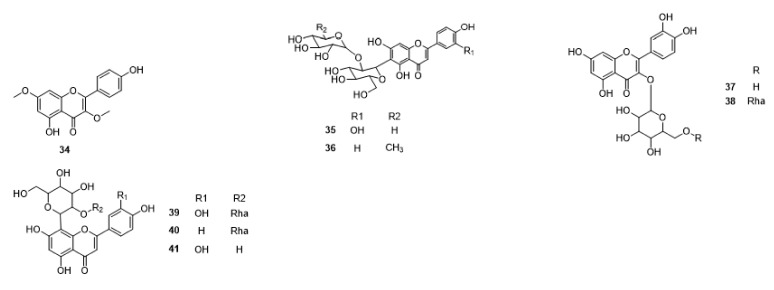

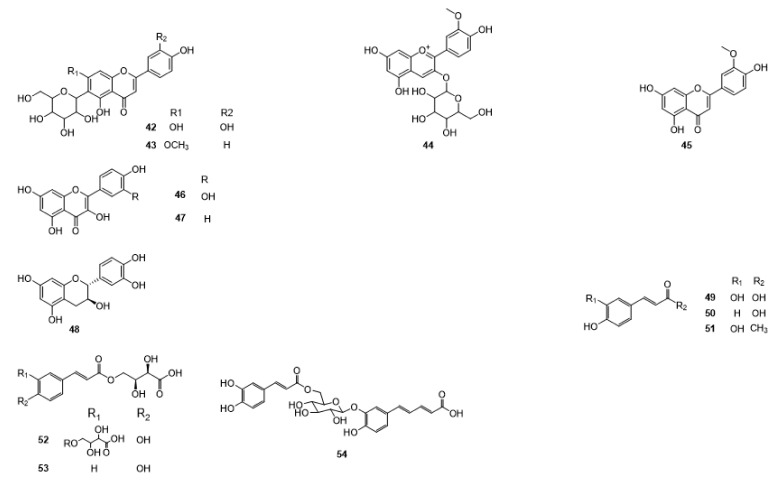

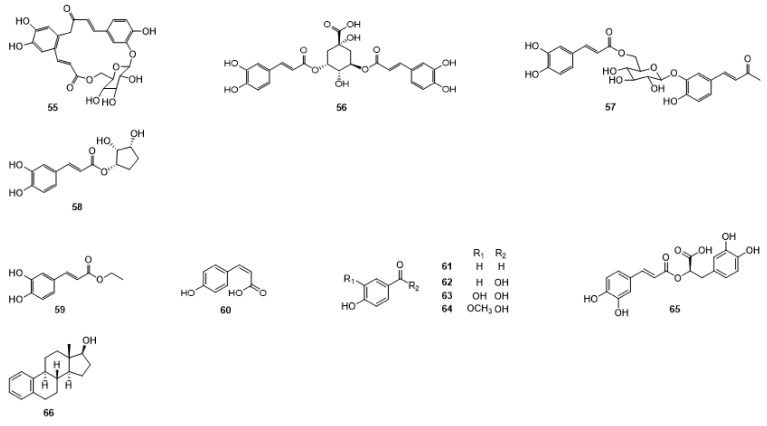

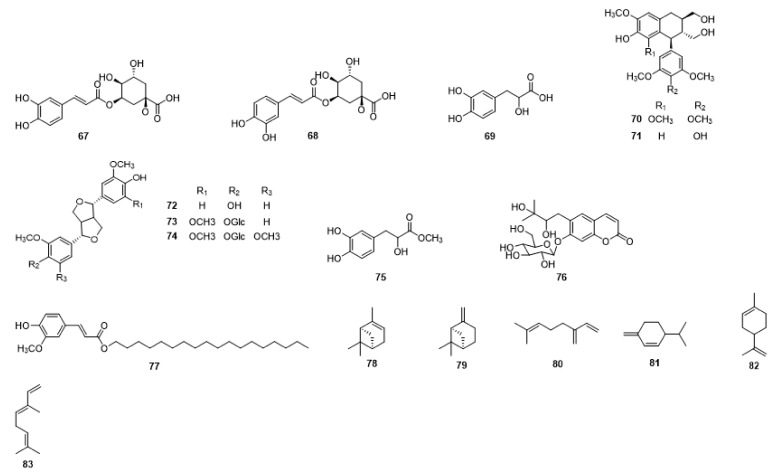

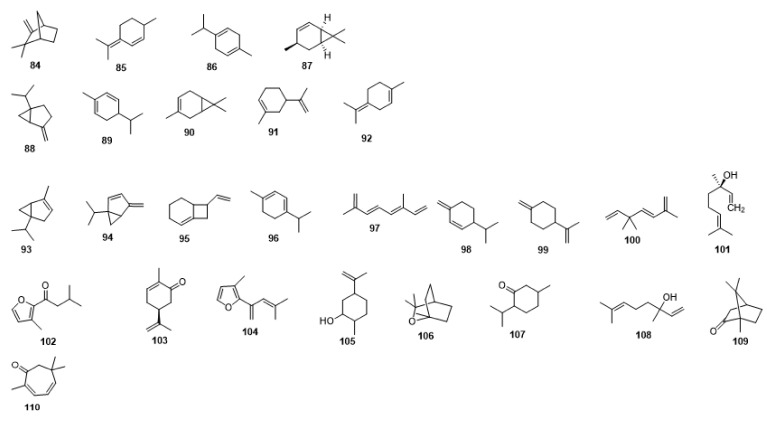

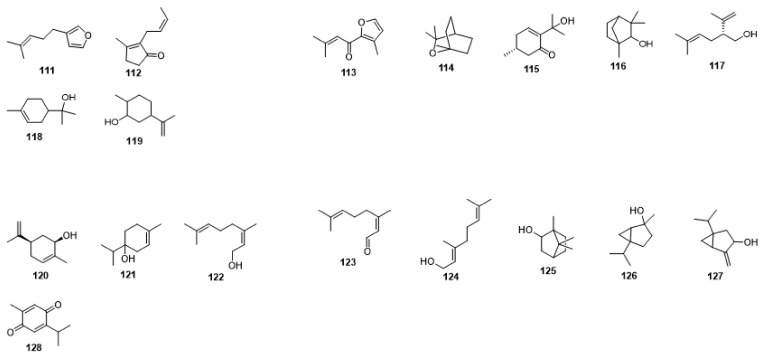

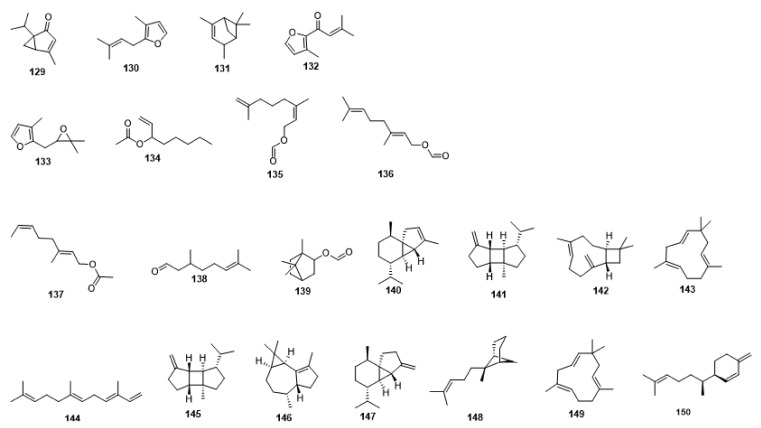

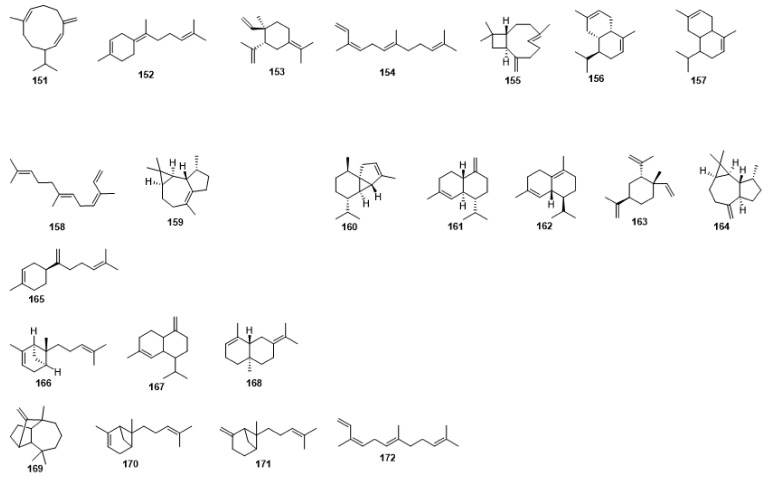

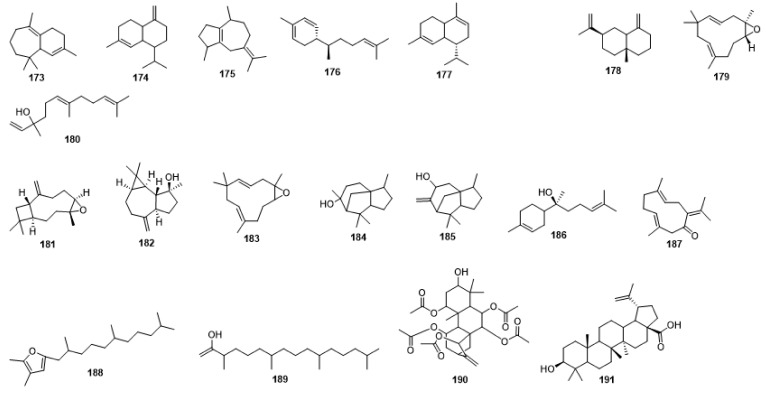

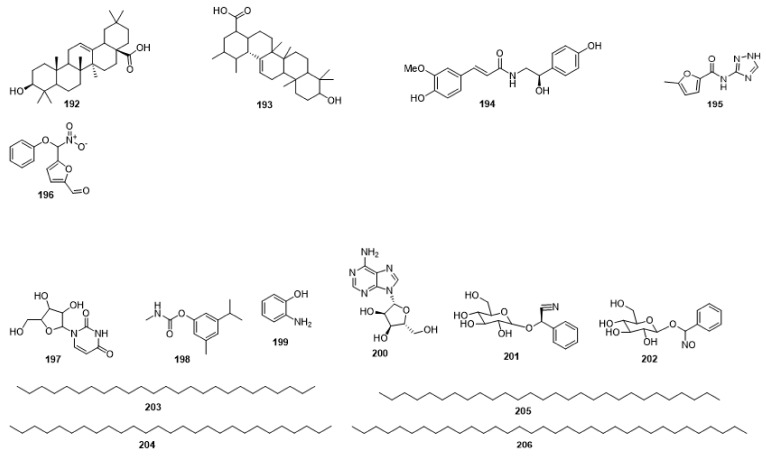

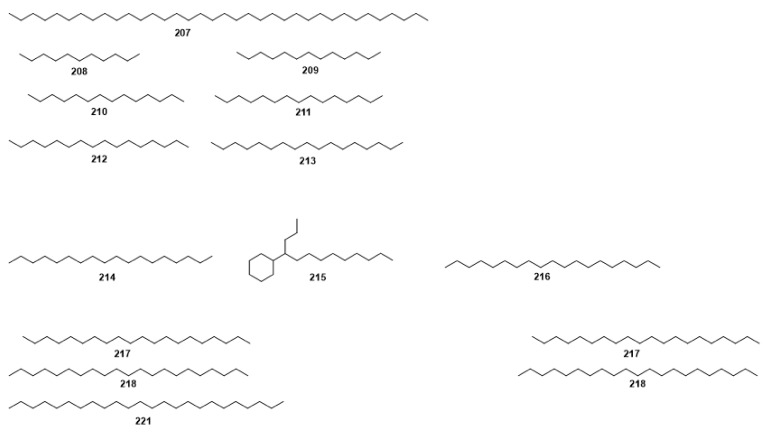

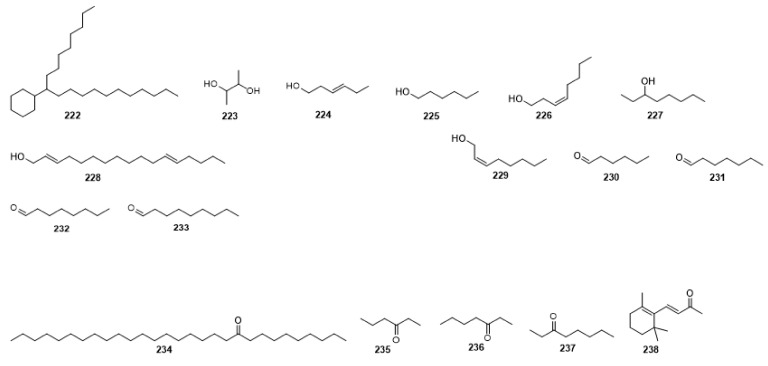

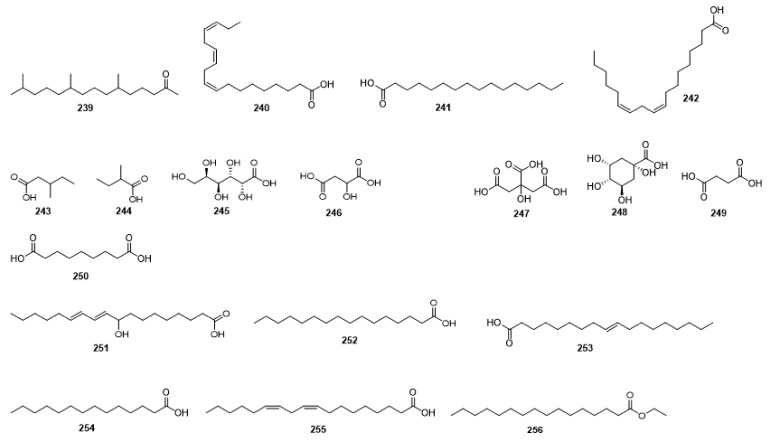

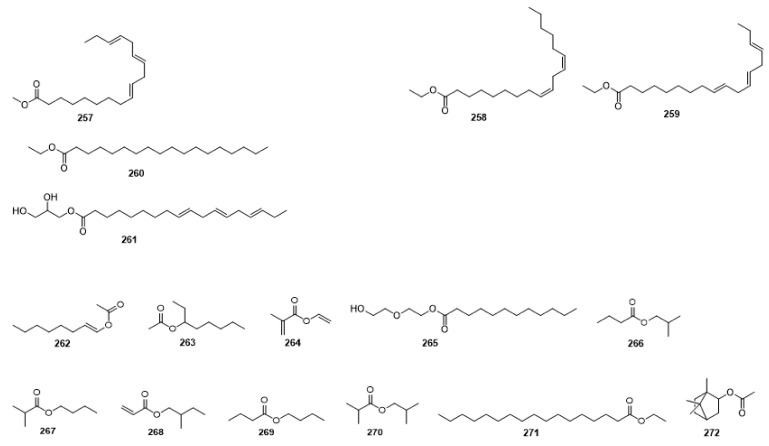

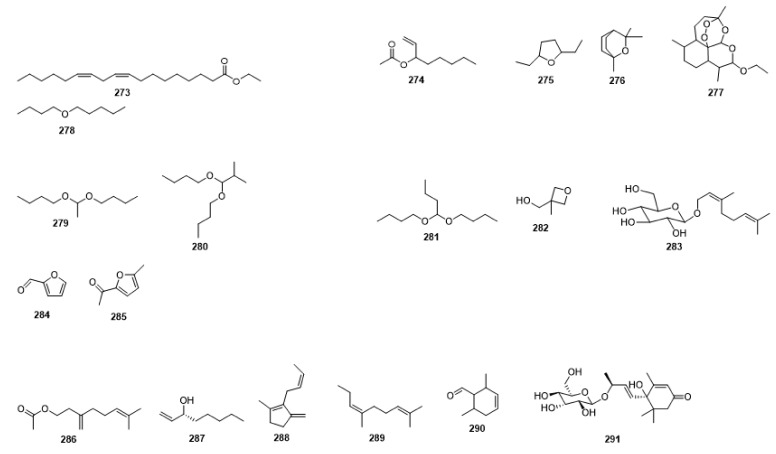

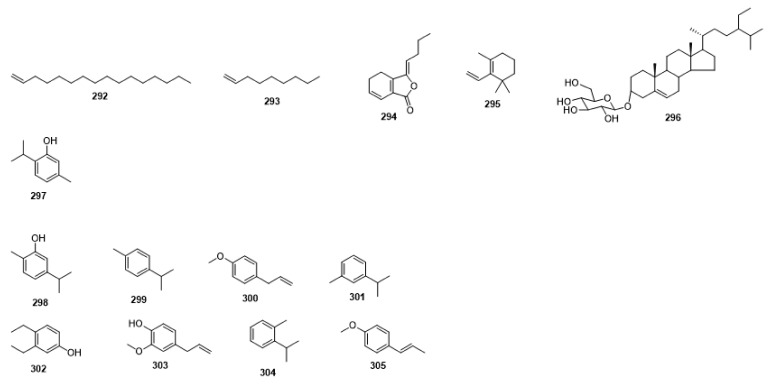

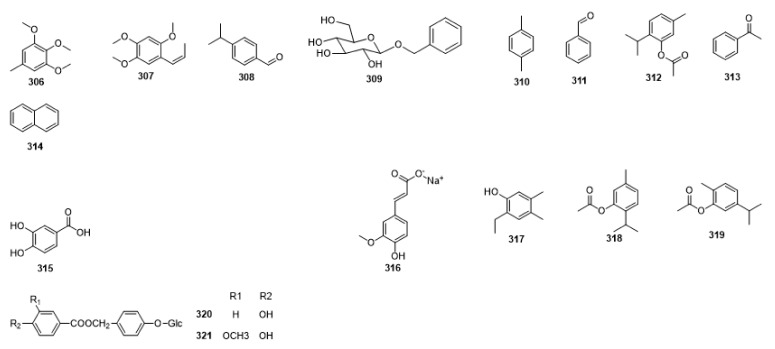

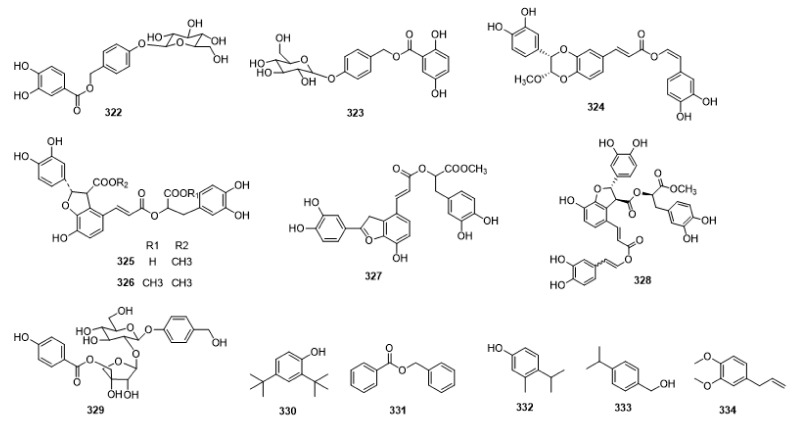

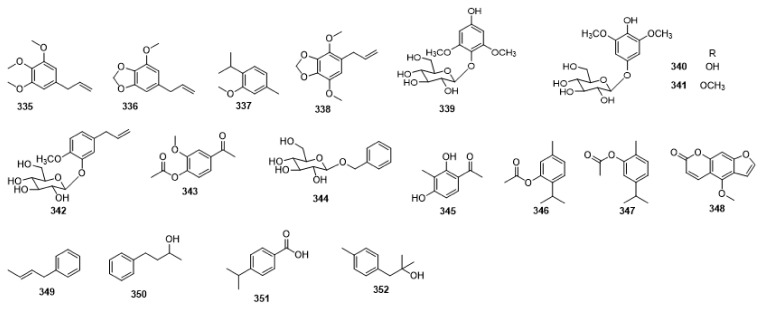



## 4. Conclusions and Prospects

This paper summarizes pharmacological activities of *E. ciliata*, among which antioxidant, anti-inflammatory, antimicrobial and insecticidal activities are the main activities, but also has antiviral, hypolipidemic, hypoglycemic, anti-tumor activities. Hence, **352** kinds of chemical constituents identified from *E. ciliata* were summarized. According to their structure types, they can be divided into flavonoids, phenylpropanoids, terpenoids, alkaloids and other compounds.

According to the existing pharmacological experiment results in vivo and in vitro, *E. ciliata* dichloromethane extract, ethyl acetate extract and essential oil all show good pharmacological activity. Carvacrol contained in *E. ciliata* is the main active ingredient of antibacterial. At present, researches on pharmacological activity of *E. ciliata* mainly focus on essential oil, and some researches involve *E. ciliata* alcohol extract, water extract and polysaccharide, but there are relatively few researches on pharmacological activities of *E. ciliata* such as analgesia, immune regulation, hypoglycemia and hypolipemia. Whether *E. ciliata* has potential pharmacological activity still needs further test to prove. The safety study of clinical dose also deserves extensive attention. 

Some representative action mechanisms of *E. ciliata* were briefly illustrated for the sake of reference. The possible action process is shown in the figure. The mitogen-activated protein kinases (MAPKs) signaling pathway chain consists of three protein kinases, MAP3K–MAP2K–MAPK, which transmit upstream signals to downstream responsive molecules through sequential phosphorylation. MAPK includes four subfamilies: ERK, p38, JNK, and ERK5. MAPK activity is thought to be regulated by the diphosphate sites in the amino acid sequence of the active ring. The active ring contains a characteristic threonine-x-tyrosine (T-x-Y) motif. Mitogen activated protein (MAP) kinase phosphorylates on two amino acid residues, thereby activating MAPK pathway. MAP kinase phosphatase (MKP) can hydrolyze phosphorylated products and inactivate MAPK pathway. The extract inhibited the activation of MAPK signaling pathway by blocking the phosphorylation of p38, JNK and ERK [18]. When stimulated, tissue cells release arachidonic acid (AA). Cyclooxygenase (COX) catalyze AA to produce a series of bioactive substances such as prostaglandins (PGs), causing inflammation. The extract can affect the COX-2 pathway by affecting the release levels of TNF-*α*, IL-6, and PGE2, which are key mediators released by macrophages during bacterial infection, so as to achieve the purpose of anti-inflammatory [4]. Carvacrol can significantly inhibit the mRNA expression of toll like receptor 7 (TLR7), interleukin-1 receptor associated kinase (IRAK4), TNF receptor associated factor (TRAF6), induced pluripotent stem-I (IPS-I), and interferon regulatory factor 3 (IRF3) in mice, thereby affecting the immunomodulatory signaling pathways of TLR7/RLR and playing an anti-H1N1 influenza virus role [19]. With the deepening of various studies, the gradual clarification of the mechanism of action has created conditions for drugs to play a better role. Two representative mechanisms of action, MAPK and COX-2, are shown in Figure 2 and Figure 3.

*E. ciliata* has rich resources and low requirements for growth environment. It can be planted artificially and has a short growth cycle. The rich essential oil content makes it possible for *E. ciliata* to be used as flavor and food additive. Undoubtedly, the development of *E. ciliata* in new dosage forms and the application to medicine, food, and other fields will provide broad development prospects in the future.

**Figure 2 molecules-27-06411-f002:**
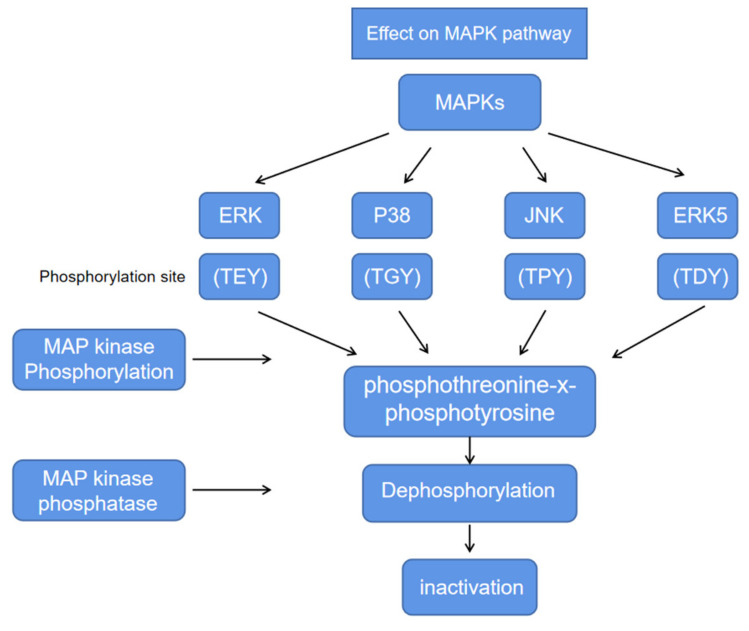
Effect on Mitogen-activated protein kinases pathway.

**Figure 3 molecules-27-06411-f003:**
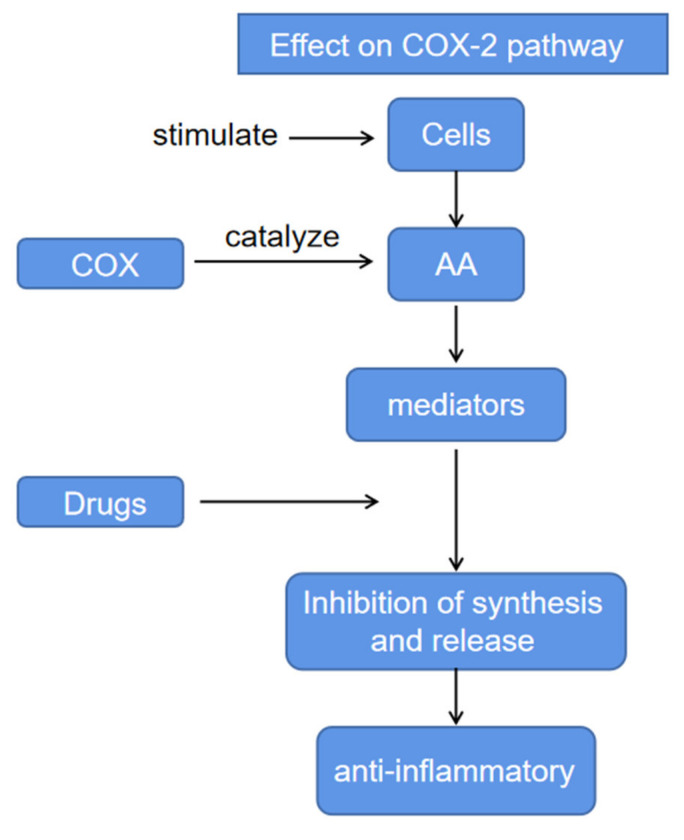
Effect on Cyclooxygenase-2 pathway.

## Data Availability

The data presented in this study are available on request from the corresponding author.

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
