# Peer review of "Elsholtzia ciliata (Thunb.) Hyland: A Review of Phytochemistry and Pharmacology"

_molecules, 2022, doi:10.3390/molecules27196411_

Round 1

Reviewer 1 Report

The manuscript entitled „Elsholtzia ciliate (Thunb.) Hyland: A review of phytochemistry and pharmacology” reviews pharmacological properties of E.ciliata plant and presents the complete list of names and structures of all 352 identified chemical compounds in that plant, which is definitely a strong point of that paper. However the manuscript is not acceptable for publication in its present form, a number of revisions is required.

Major remarks:

·         The first two sentences of Introduction suggest that E.ciliata and Mosla chinensis is the same plant since it is written that “Elsholtzia ciliate (Thunb.) Hyland….. It is composed of aerial parts of Mosla chinensis….”. this is rather confusing since Elsholtzia and Mosla are two different and accepted genera of Lamiaceae family. Please correct it or describe it clearly.  

·         The beginning parts of the manuscript, namely Introduction and paragraphs 1, 2 and 2.1 are very hard to read. The thorough English proof reading is required here. There are also numerous stylistics mistakes. See some suggestions in specific remarks.

·         The E.ciliata contains many chemical components belonging to different chemical classes, which all are presented in paragraph 1 chemical constituents referring also to the table at the end. Since the chemical classes are characterized with different chemical properties and structures, there should be added a brief summary of each chemical class containing chemical and structural properties with a pre-cursors indication.

Specific remarks:

It should be pointed out that the manuscript version for peer review without line numbers doesn’t make the review process easy. Therefore it is only possible to refer to the text using only paragraph numbers.

·         E.ciliata or Elsholtzia ciliata seems to be correct name, please correct it through the manuscript, also in the title.

·         Abstract: “pharmacological activities” better replace with pharmacological properties

·         Abstract: “there are many researches on essential oil” better replace with there are many researches using essential oil

·         Introduction: The part describing JXR plants appearance should be rewritten using full sentences containing verbs.

·         Introduction: “E.ciliate is a herbaceous plant which distributed” better “E.ciliata is a herbaceous plant which is distributed”   

·         Introduction: “, Europe and North America are also introduced and cultivated” is mistaken maybe better with “,  while in Europe and North America it was also introduced and cultivated”

·         Introduction: “dispelling exegonous evils” not necessary to mention that, maybe change with more scientific term.

·         Paragraph 1: “The chemical components of E.ciliate mainly include volatile components and non-volatile component” The sentence doesn’t introduce any information specific for E.ciliata because this is true for many plants. Add more information or rephrase.

·         Paragraph 2: The first sentence is a repetition of the last part of Introduction, which is not far above. This should be rewritten.

·         Paragraph 2.1: “according to free radical scavenging experiment of H.X.P., the result showed that E.ciliata extract DPPH (…) and ABTS (…), which had certain antioxidant ability” The sentence is completely wrong, some parts are missing. DPPH and ABTS are free radicals and this sentence suggests that they have antioxidant ability. This must be corrected. Between words “extract” and “DPPH” should also be something added because now it doesn’t make any sense.

·         Paragraph 2.6: “The hypidemic activity” replace with “The hypolipidemic activity”

·         Paragraph 3: “they can be divided into flavonoids, phenylpropanoids……. and other compounds, mainly flavonoids and terpenes” The second flavonoids in that sentence should be replaced.  

Reviewer 2 Report

Elsholtzia ciliate (Thunb.) Hyland: A review of phytochemistry and pharmacology

1.      In title the name of the plant is written as Elsholtzia ciliate, while in other places it is written as Elsholtzia ciliata. Which one is correct?

2.      The abstract must clearly show what incited the researchers for review

3.      Typographic/spelling mistakes must be corrected throughout the paper. Some are highlighted (pdf attached).

4.      References have been added at the start of proceeding sentence, must be rectified.

5.      English language need improvement

6.      Many terms have been written as abbreviated without first writing in expanded form. Such as MP-1, TGF, PGE2.. Rectify this.

7.      Page 3 …it can be seen that E. ciliata has the potential to prevent diseases caused by excess free radicals….The diseases must be enumerated in one sentence

8.      Page 3. The fraction E DPPH? Explain

9.      The units of butylated hydroxytoluene (0.45), butylated hydroxyanisole (0.21) and Vc (0.41), are missing. What is Vc?

1.  Re-write the sentence “E.ciliata showed well activity in insecticidal aspect”. Page 5, 1st line

11.  Thymol, carvacrol and β -thymolcontained in JXR essential oil…. Space … page 5

12.  The discussion section needs more explanation.

Reviewer 3 Report

The manuscript has some minor concerns:

1) Expand the abstract sections. 

2) expand the discussion part.

3) Put conclusion in a separate section.

4) Include a figure illustrating the biological potential of the plant along with the possible mechanisms of action

5) Give a note on traditional potential of the plant just after the introduction section

Round 2

Reviewer 1 Report

The paper now is improved, the Authors replied to almost all comments except one regarding addition of descriptions of presented  compound classes identified in E. ciliata to paragraph 1. I still think that descriptions should be added there.

Moreover, there are few corrections to make:

Please change corrected sentence from paragraph 2.1: “According to free radical scavenging experiment of Huynh Xuan Phong, the rusult showed that E.ciliata extract had certain scavenging ability against 2,2-diphenyl-1-picrylhydrazyl (DPPH) and 2,2' -azino-bis (3-ethylbenzothiazoline -6-sulfonic acid) (ABTS), with IC50 values of 495.80 ± 17.16 and 73. 59 ± 3.18 mg/mL,which had certain antioxidant ability” to  According to free radical scavenging experiment of Huynh Xuan Phong, the rusult showed that E.ciliata extract had certain scavenging ability against 2,2-diphenyl-1-picrylhydrazyl (DPPH) and 2,2' -azino-bis (3-ethylbenzothiazoline -6-sulfonic acid) (ABTS), with IC50 values of 495.80 ± 17.16 and 73. 59 ± 3.18 mg/mL.”

The figure captions should be more concrete.

There is lack of accurate reference to figures both 1 and 2 in the paragraph 3. Please change that.

Reviewer 2 Report

The authors have done the suggested changes. I recommend the article for publication.

Author Response

Has been further modified.